# Cohort profile: St. Michael's Hospital Tuberculosis Database (SMH-TB), a retrospective cohort of electronic health record data and variables extracted using natural language processing

David Landsman[1], Ahmed Abdelbasit[2], Christine Wang[2], Michael Guerzhoy[3,4,5], Ujash Joshi[4], Shaun Mathew[6], Chloe Pou-Prom[7], David Dai[7], Victoria Pequegnat[8], Joshua Murray[7], Kamalprit Chokar[9], Michaelia Banning[7], Muhammad Mamdani[2,4,5,7,10,11,12,13], Sharmistha Mishra[1,2,11,13⊚], Jane Batt[2,13,14⊚]*

1 MAP Centre for Urban Health Solutions, Li Ka Shing Knowledge Institute, St. Michael's Hospital, Unity Health Toronto, Toronto, Ontario, Canada, 2 Department of Medicine, University of Toronto, Toronto, Ontario, Canada, 3 Princeton University, Princeton, New Jersey, United States of America, 4 University of Toronto, Toronto, Ontario, Canada, 5 Li Ka Shing Knowledge Institute, St. Michael's Hospital, Unity Health Toronto, Toronto, Ontario, Canada, 6 Department of Computer Science, Ryerson University, Toronto, Ontario, Canada, 7 Unity Health Toronto, Toronto, Ontario, Canada, 8 Decision Support Services, St. Michael's Hospital, Unity Health Toronto, Toronto, Ontario, Canada, 9 Division of Respirology, Department of Medicine, St. Michael's Hospital, Unity Health Toronto, Toronto, Ontario, Canada, 10 Leslie Dan Faculty of Pharmacy, University of Toronto, Canada, Toronto, Ontario, Canada, 11 Institute of Health Policy, Management, and Evaluation, University of Toronto, Toronto, Ontario, Canada, 12 Vector Institute, Toronto, Ontario, Canada, 13 Institute of Medical Science, University of Toronto, Toronto, Ontario, Canada, 14 Keenan Research Center for Biomedical Science, St. Michael's Hospital, Unity Health Toronto, Toronto, Ontario, Canada

⊚ These authors contributed equally to this work.
* Jane.batt@utoronto.ca

**Data Availability Statement:** The validated NLP rulesets are publicly available for use from: https://

## Abstract

### Background

Tuberculosis (TB) is a major cause of death worldwide. TB research draws heavily on clinical cohorts which can be generated using electronic health records (EHR), but granular information extracted from unstructured EHR data is limited. The St. Michael's Hospital TB database (SMH-TB) was established to address gaps in EHR-derived TB clinical cohorts and provide researchers and clinicians with detailed, granular data related to TB management and treatment.

### Methods

We collected and validated multiple layers of EHR data from the TB outpatient clinic at St. Michael's Hospital, Toronto, Ontario, Canada to generate the SMH-TB database. SMH-TB contains structured data directly from the EHR, and variables generated using natural language processing (NLP) by extracting relevant information from free-text within clinic, radiology, and other notes. NLP performance was assessed using recall, precision and $F_1$

github.com/mishra-lab/tb-nlp-rulesets. Data collected in SMH-TB contains sensitive patient information and as such, researchers interested in conducting TB-related research using the data are welcome to contact the Unity Health Toronto Research Ethics Board (researchethics@smh.ca) and the corresponding author and submit a request. The study team welcomes collaboration and use of the database, and all external requests will be screened to ensure adequate data exists to enable a collaboration. The project will then undergo the approval process of the Research Ethics Board of Unity Health Toronto. Data provided to researchers can either be the de-identified version of the SMH-TB database, or the full identifiable version, based on their research needs and REB approval.

**Funding:** Supported by the Ontario Early Researcher Award Number ER17-13-043 (to SM). The funders had no role in study design, data collection and analysis, decision to publish, or preparation of the manuscript.

**Competing interests:** The authors have declared that no competing interests exist.

score averaged across variable labels. We present characteristics of the cohort population using binomial proportions and 95% confidence intervals (CI), with and without adjusting for NLP misclassification errors.

## Results

SMH-TB currently contains retrospective patient data spanning 2011 to 2018, for a total of 3298 patients (N = 3237 with at least 1 associated dictation). Performance of TB diagnosis and medication NLP rulesets surpasses 93% in recall, precision and $F_1$ metrics, indicating good generalizability. We estimated 20% (95% CI: 18.4–21.2%) were diagnosed with active TB and 46% (95% CI: 43.8–47.2%) were diagnosed with latent TB. After adjusting for potential misclassification, the proportion of patients diagnosed with active and latent TB was 18% (95% CI: 16.8–19.7%) and 40% (95% CI: 37.8–41.6%) respectively

## Conclusion

SMH-TB is a unique database that includes a breadth of structured data derived from structured and unstructured EHR data by using NLP rulesets. The data are available for a variety of research applications, such as clinical epidemiology, quality improvement and mathematical modeling studies.

## Introduction

Tuberculosis (TB) is the top infectious killer worldwide, resulting in 1.6 million deaths in 2017 [1]. 1.7 billion people carry the latent form of the infection, of whom 10% at minimum, will develop the active, infectious form of disease. Latent TB infection (LTBI) progression to active disease can be prevented and TB can be cured, with appropriate antibiotics taken over many months. TB is endemic in many low-income countries and particularly prevalent in Asia and Africa. The World Health Organization recommends the treatment of LTBI as part of the global "End TB Strategy", and an achievable goal critical to TB elimination in high-income countries [2, 3].

Given the burden of active TB disease is disproportionately carried in low-resource settings, research addressing disease epidemiology, treatment (including clinical trials and programs of delivery), and the use and utility of innovative and point of care diagnostics is often completed in the populations of countries with highest burden of TB. The prevalence of LTBI on the other hand, is considerable even in high-income countries (CDC estimates 13,000,000 people living the USA have LTBI [4]) and thus research ranging from basic pathogenesis to program development can be conducted on the global population. Indeed while advances in biomedical research over the past 1 to 2 decades have delivered successes ranging from rapid point-of-care diagnostics testing for pulmonary TB to the development of novel therapeutics such as bedaquiline and delamanid, many questions remain, including, for example, discovering biomarkers that precisely indicate individuals at risk of LTBI activation and developing programs of TB care that ensure efficacy, are equitable and resilient [1, 5].

Many primary care practices and hospitals in high-income countries have curated electronic health record (EHR) data for research and surveillance [6–9], that improve ease of access to information and data sharing for collaborative work. The use of EHRs in hospital and office-based clinical practices has risen substantially in the past decade, providing rich

data sources that have the potential to simultaneously improve patient care and advance research initiatives [10, 11]. Most EHR-derived databases are however limited to structured data, such as demographic information collected at patient registration, laboratory tests and results and diagnostic codes used in physician billing. As such, the rich, granular data embedded within unstructured (text) data from dictated notes on both hospital admitted and clinic patients are excluded [12, 13] unless these variables are abstracted via manual chart review [14, 15] or natural language processing (NLP) [16–18].

The intention of NLP is to "develop computational models for understanding natural language" [19]. NLP algorithms extract information (e.g. change unstructured to structured text), perform syntactic processing (e.g. sentence detection) capture meaning (e.g. assign concepts to words) and detect relationships between concepts [20]. They range from simple rule-based approaches to statistical and machine learning models. Although there has been an exponential rise in publications citing the use of EHR in clinical and translational research over the last decade, the concurrent uptake in the application of NLP methodologies to extract information in clinical studies [21, 22] has remained more limited [18].

Here we develop and describe a digital retrospective clinical database that combines structured data, unstructured (text) data, and variables derived from transforming unstructured data to structured data using natural language rulesets, among patients assessed in an inner-city outpatient TB clinic at St Michaels Hospital (SMH) of Unity Health Toronto in Toronto, Ontario, Canada. Approximately 2000 people (5.6 per 100,000 people) are diagnosed with active TB in Canada [23] annually and 1.3 million are estimated to have LTBI. The SMH TB clinic cares exclusively for individuals with suspected or diagnosed active TB and LTBI, seeing 1800–2200 patient-visits each year, and assessing and developing a diagnostic and management plan for 670–800 new patients each year. The SMH-TB database aims to be a resource for scientists who are conducting research into many facets of TB, ranging from observational epidemiology to emulated trials and quality improvement and implementation science research.

The purpose of this profile is to describe our methodology, present the cohort and SMH-TB database validation. Access to the database is available to collaborators wishing to work with the research team of the SMH TB clinic. The NLP rulesets developed to extract variables from the unstructured data in the EHR are publicly available on GitHub [24].

## Materials and methods

### Cohort description

The database compiles all data available on all TB clinic patients (N = 3298) treated at SMH from April 2011 to December 2018. The database contains socio-demographic information surrounding immigration, housing status, insurance, and clinical information including laboratory and imaging results, comorbidities, diagnoses and treatment. Ethics approval for development and validation of the database was obtained from the Unity Health Toronto Research Ethics Board (REB 19–080). Patient consent was not required or obtained as per the Tri-Council Policy Statement 2 (TCPS2), since only retrospective data were collected from clinical charts [25].

Patients are referred to the TB outpatient clinic predominantly from Public Health Units in the Greater Toronto area (population of 6 million), Canada Immigration and Citizenship, Occupational Health and Safety Departments of Toronto area hospitals, community health care professionals (physicians, nurse-practitioners), and SMH staff physicians caring for an admitted patient or a patient in the emergency room (ER). When including a patient in our database we consider all available encounters, including inpatient admissions and ER visits.

### Data collection

St. Michael's Hospital EHR is managed by several systems. The Enterprise Data Warehouse (EDW) stores and manages structured data including patient demographics and medical test results. Soarian stores the unstructured patient data, which includes dictated clinical notes. SMH-TB retrieved data of patients registered and assessed in the TB outpatient clinic to provide a comprehensive description of patient characteristics, disease, management and clinical trajectory. SMH-TB is restricted to a start-date of April 2011, which is the date of initiation of EHR at SMH. Fig 1 shows the data flow and data sources for the SMH-TB database.

The SMH-TB database stores patient characteristics and encounter data in separate tables, which can be linked together using unique, de-identified patient or encounter IDs. Fig 2 presents the tables provided in SMH-TB, and the granularity of the data they contain.

A detailed collection of all the variables available in the database is provided in Table 1.

**Removing identifiable information.** There are two versions of SMH-TB. The full version includes indelible patient identifiers such as a patient's provincial health insurance (Ontario Health Insurance Plan) number; their SMH-specific medical record number; all patient encounters whose encounter record is specific to a given patient; laboratory test records whose encounter record is also specific to a given patient; and all unstructured text data per encounter per patient. The patient identifiers allow for a fully linked database, which can be updated and linked via future data extraction. The identifiable unstructured data are also retained to support the development and testing of additional natural language rulesets.

The de-identified version of SMH-TB is the version that will be primarily used for research studies. It excludes the unstructured data and has been stripped of the following: hospital patient ID, hospital encounter ID, address and day and month of date of birth. Each patient

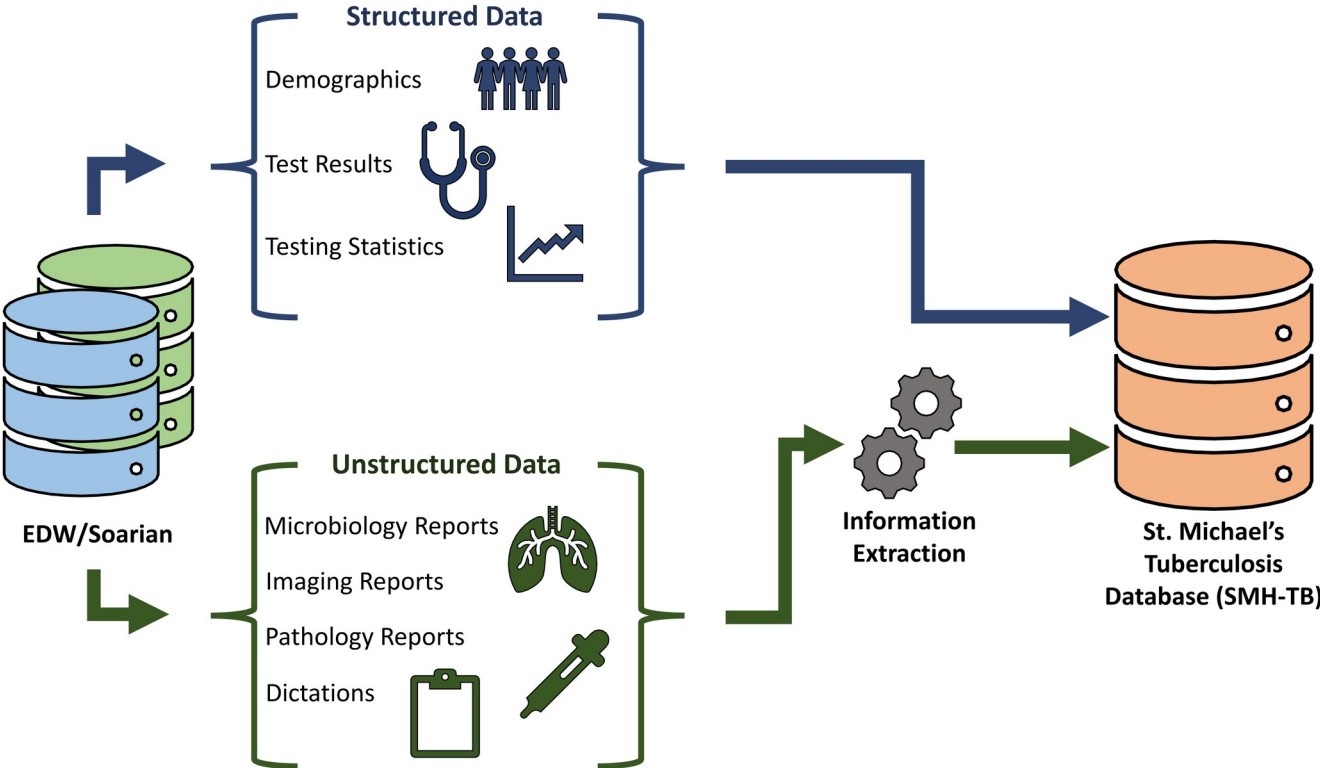

**Fig 1. Data sources for SMH-TB database.**

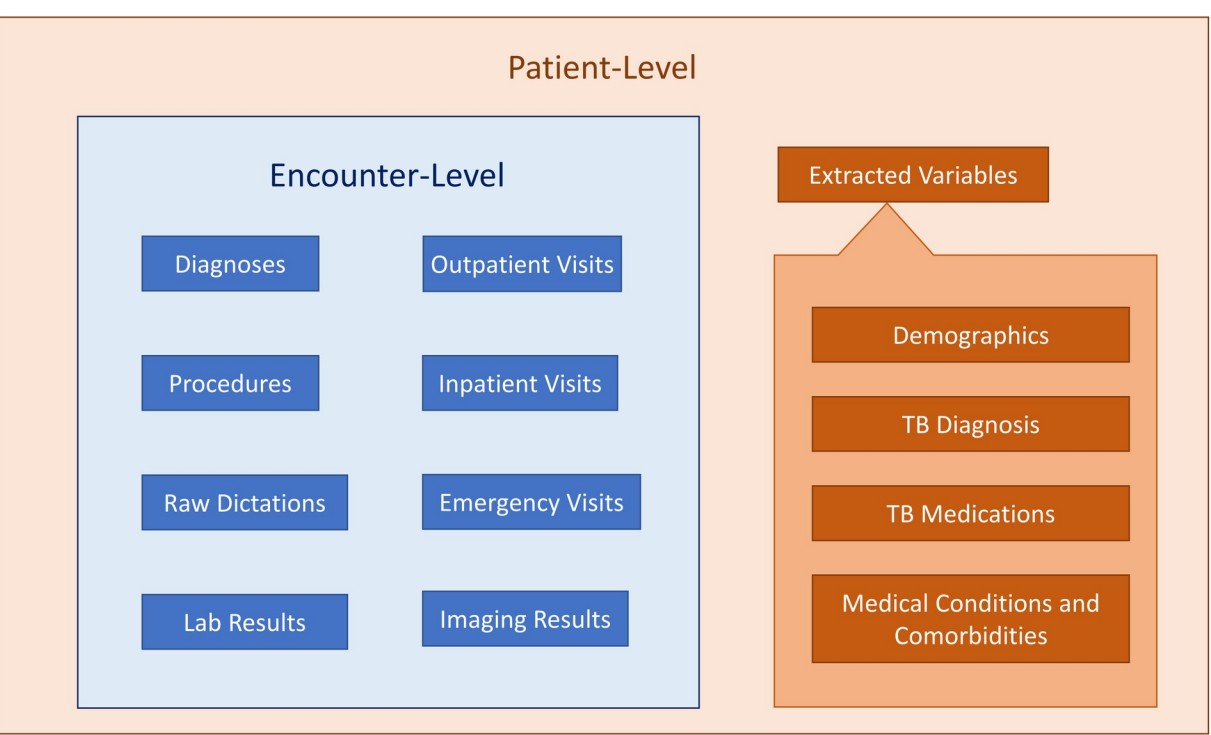

**Fig 2. Patient-level and encounter-level data in SMH-TB.**

**Table 1. Variables available in SMH-TB from both structured and unstructured sources.**

| Demographics | Tuberculosis Diagnosis |
|---|---|
| Patient ID | Known TB exposure* |
| MRN | BCG vaccination status* |
| Sex | TST performed* |
| Date of birth | TST induration* |
| Street address | TST interpretation* |
| Postal code[a] | IGRA performed* |
| Country of origin* | IGRA interpretation* |
| Year of immigration* | Diagnosis of active TB* |
| Immigration status | Diagnosis of LTBI* |
| Housing status | |
| Insurance status | **Tuberculosis Medications** |
| Patient is a healthcare worker* | Ever started isoniazid* |
| | Ever started rifampin* |
| **Encounter Details** | Ever started pyrazinamide* |
| Encounter ID | Ever started ethionamide* |
| Encounter type | Ever started vitamin B6* |
| Encounter date | |
| Direct cost[b] | **Medical Conditions and Comorbidities** |
| Indirect cost[c] | Autoimmune conditions[d]* |
| | Diabetes* |
| **Aggregate Variables** | Hematological malignancy* |
| Number of sputum inductions | Non-hematological malignancy* |

(*Continued*)

**Table 1.** (Continued)

| Demographics | Tuberculosis Diagnosis |
|---|---|
| Number of chest x-rays | Transplant performed* |
| Number of chest computed tomography | Renal failure[e]* |
| Hospital admission during course of TB outpatient care | Silicosis* |
| Number of emergency room visits during course of TB outpatient care | Hepatitis B |
| | Hepatitis C |
| **Laboratory Results** | HIV status* |
| AST | |
| ALT | **Microbiology Reports**** |
| CBC (Hb, Platelets, WBC) | **Radiology Reports**** |
| Cr | **Pathology Reports**** |
| Bilirubin | |

MRN: Medical record number; AST: Aspartate transaminase; ALT: Alanine transaminase; CBC: Complete blood count; Hb: Hemoglobin; WBC: White blood cells; Cr: Creatinine; TB: Tuberculosis; BCG: Bacillus Calmette–Guérin; TST: Tuberculin sensitivity test; IGRA: Interferon gamma release assay; LTBI: Latent tuberculosis infection; HIV: Human immunodeficiency viruses

[a]The database only stores the Forward Sortation Area portion of the postal code of the patient's residence.

[b]Direct cost corresponds to health care services directly associated with the patient's care including all nursing, allied health, diagnostic and therapeutic services, pharmaceutical and medical/surgical supplies for each visit.

[c]Indirect cost corresponds to administrative and support services performed on behalf of all patients including information system and housekeeping overheads.

[d]Autoimmune conditions include: Sjogren's syndrome, arthropathy, spondyloarthropathy, psoriatic arthritis, rheumatoid arthritis, reactive arthritis, mixed connective tissue disease, connective tissue disease, systemic lupus erythematosus, CREST syndrome, dermatomyositis, Wegener's granulomatosis, Goodpasture syndrome, vasculitis and psoriasis.

[e]Renal failure includes: nephropathy, renal insufficiency and glomerulonephritis.

*Variables collected from unstructured dictations and reports using natural language rulesets

**Unstructured text from which variables will be generated using natural language rulesets

and encounter are then re-coded with new unique IDs, and with the age in years on the date of the first TB clinic encounter.

**Patient identification and validation.** The Decision Support Services (DSS) at SMH identified encounters which were coded as services provided in the TB outpatient clinic to identify all TB patients. We then randomly selected a list of 200 patients seen in the TB outpatient clinic (using clinic schedules with unique patient identifiers stored separately from the EDW) to manually validate the codes used by DSS to identify TB clinic outpatients, and validated that all (100%) identified patients were registered in the TB clinic. To ensure high specificity of our identification of TB clinic patients, we examined additional metadata (such as a mention of the TB clinic in the patient's dictations) and removed patients without matching metadata. SMH-TB therefore may include the rare patient where the clinic visit codes in the EDW erroneously labelled a visit as a TB clinic visit, but this estimate is expected to be <0.2% because of the additional metadata checks. The hospital unique patient identifier for each individual was then cross referenced to lists of all individuals with inpatient stays and ER visits to derive TB patient data from all sites of contact for TB care.

**Data transformation (unstructured text to structured variables).** Unstructured clinician dictations were used to create patient-level variables on demographics, TB diagnosis, TB medications and comorbidities. The data for these variables were extracted using rule-based

information extraction tool CHARTextract [26]. CHARTextract uses regular expressions in order to perform pattern matching on text. Regular expressions have been used to perform data extraction and even classification due to their high expressivity [17, 27, 28]. These capabilities come at the cost of a complex syntax, and thus rule creation typically involves the expertise of a clinician who understands the subject matter and an interpreter who can express the idea into regular expression syntax. We created a tiered rule system, where primary rules are used to filter text at the sentence level using a scoring system and secondary rules can be used to further enhance the weighting of the sentence. The tool applies the user-created rules to the data and extracts the variables on-the-fly. The interface displays mismatches between the tool prediction and the gold-standard label. Users can iterate on the rule creation process, allowing for easy refinement and quick development of the rules. Fig 3 shows a component of a ruleset for extracting diagnosis of active tuberculosis.

In order to create the rulesets used by CHARTextract, two clinicians (JB, SM) from the TB outpatient clinic were consulted on dictation language and style. Clinicians (JB, SM, AA, and KC) and a medical student (CW) served as chart abstractors and manually labeled dictations for 200 patients from a subset of the dataset to be used for validation. The set of 200 patients was selected from consecutive clinic visits based on registered patient lists external to the EHR. Data abstraction was done using the QuickLabel tool which provides a user interface for streamlined labelling of specific variables, as well as the option to label multiple variables simultaneously (Fig 4) [29].

Chart abstractors were provided a priori instructions on how to interpret the text to identify the value labels (e.g. Yes, No, Indeterminate, Not recorded) for variables in QuickLabel. Abstractors additionally made notes on novel/unanticipated dictation wording and phrasing and met with programmers to assist with iterative refinement of the natural language ruleset. Table 2 shows the dictated text wording and phrasing defining the value labels for the comorbidity diabetes mellitus, as an example variable. Varied wording and phrasing were encountered in the dictated clinic notes and with which the value labels of "yes" or "no" for diabetes mellitus were assigned. Natural language rulesets were then written based on these dictation styles.

Refinement of the natural language rulesets was done by comparing the labels extracted by the rulesets via CHARTextract with the manual labels on the training dataset. The refined rulesets are available as a real-time source as additional variables from unstructured data (microbiology, radiology, and pathology reports) are generated [24].

**Evaluation of data extraction.**    To measure the performance of our rulesets and evaluate their generalizability to unseen data, we calculated recall, precision and $F_1$ scores. Recall (sensitivity) measures the ability of the classifier to correctly distinguish true positive from false negative examples. Precision (positive predictive value) measures the ability of the classifier to correctly distinguish true positive from false positive examples. The $F_1$ score computes a harmonic mean of precision and recall. Recall, precision and $F_1$ score were averaged across variable labels.

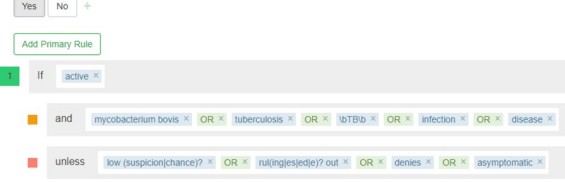

**Fig 3. Example of a component of a ruleset for extracting a variable (active TB diagnosis) from unstructured text in clinical dictations (using CHARTextract).**

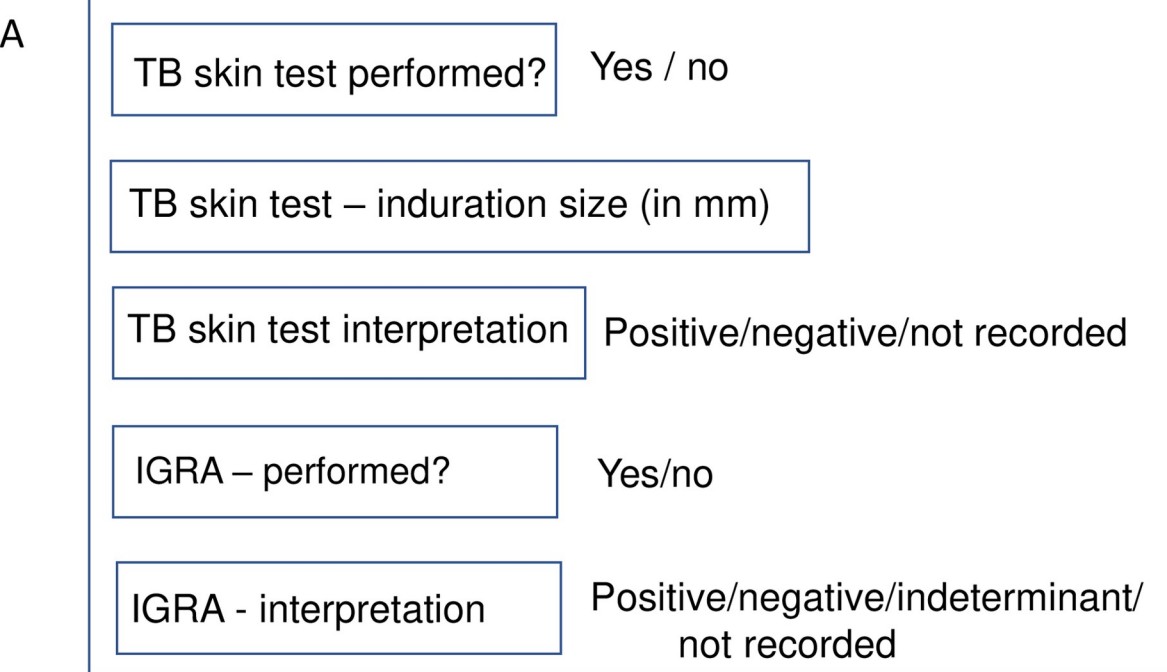

Fig 4. QuickLabel interface for manual variable abstraction. **(A)** Value labels are shown for example variables—the Tuberculin Skin Test (TST) and Interferon Gamma Release Assay (IGRA). **(B)** A screen shot of a representative data extraction using the Quicklabel tool. The corresponding sentences containing the variables of interest are highlighted in yellow.

**Table 2. Derivation of the value labels for diabetes mellitus.**

| VARIABLE: | DIABETES MELLITUS | |
|---|---|---|
| **Wording/phrasing with the disease state** | **Alternate terms for the disease state** | **Value label** |
| has a past medical history of. . . | DM | yes |
| has a history of. . . | diabetes | yes |
| has. . .. | type two diabetes | yes |
| has been diagnosed with. . .. | type one diabetes | yes |
| is on medication for . . . | | yes |
| is known to have. . .. | | yes |
| reports the following on PMH. . . | | yes |
| reports the following on past medical history | | yes |
| past medical history is significant for. . . | | yes |
| PMH is significant for . . . | | yes |
| past medical history includes. . . | | yes |
| PMH includes. . . | | yes |
| **Wording/phrasing with the disease state** | | |
| no past medical history of. . .. | | no |
| no history of . . . | | no |
| no known history of. . . | | no |
| does not have. . . | | no |
| never been diagnosed with. . . | | no |
| is not aware of a diagnosis of. . . | | no |
| has never been he/she/they has. . . | | no |
| **Wording/phrasing without the disease state** | | |
| no significant past medical history | | no |
| no significant PMH | | no |
| nothing on PMH | | no |
| nothing on past medical history | | no |
| nil on past medical history | | no |
| nil on PMH | | no |

DM: diabetes mellitus, PMH: past medical history.

**Binomial proportions estimated from extracted variables.** We used the refined rulesets to extract variables from the full dataset of patients with at least 1 dictation (N = 3237). We converted "Yes/No/Not recorded" and "Positive/Negative/Unknown/Not recorded" variables into binary 0–1 variables by assigning a value of 1 to patients with an extracted value of "Yes" or "Positive", and a value of 0 otherwise. We estimated the proportion and 95% confidence intervals of patients for which the rulesets extracted "Yes" or "Positive" for these variables using two methods: (1) logistic regression model without covariates, and (2) MC-SIMEX model that accounts for the misclassification error in the extracted variables that was calculated from the set of 200 manually abstracted patients [30]. Briefly, for a binary random variable Y, we estimate the probability P(Y = 1) using a logistic regression model without covariates, given by:

$$P(Y = 1) = h(\beta_0)$$

where h is the logistic function. Under the MC-SIMEX model, the binary random variable was

observed with misclassification errors, denoted by $Y^*$. We estimate the probability $P(Y^* = 1)$ as:

$$P(Y^* = 1) = h(\beta_0^*)$$

where $\beta_0^*$ is defined as:

$$\beta_0^*(\lambda) = h^{-1}[\pi_{11}^\lambda h(\beta_0) + (1 - \pi_{00}^\lambda)(1 - h(\beta_0))]$$

$\pi_{00}$ and $\pi_{11}$ denote the specificity and sensitivity of $Y^*$, respectively, and $\lambda$ is the misclassification parameter. The final estimate for $\beta_0^*$ is computed by a simulation-extrapolation procedure described in [30].

**Example application of the cohort–regression analysis of variables associated with LTBI treatment.** To demonstrate an application of the dataset to address clinical research questions we compared the proportion of individuals diagnosed with LTBI who received LTBI treatment based on three factors: age-group (10–40, 40–70, and 70–100 years), sex (female vs. male), and housing status (housed vs. underhoused). We used logistic regression to estimate the crude odds ratio for treatment.

## Results

### Population

A patient overview based on demographics is presented in Table 3. 3298 patients were included in the database. The median age of the patients is 45 years, with an interquartile range of 34 to 58. There is a higher percentage of females than males in the cohort, around 57%. At least 79% of the clinic patients were born outside of Canada, based on data extracted from patients' dictations. The vast majority of patients were adequately housed, with publicly funded provincial health care insurance (OHIP).

### Evaluation of data extraction

A summary of the rulesets' performance metrics for the 25 variables extracted from unstructured dictations is presented in Table 4. Diagnosis of active TB and LTBI rulesets had 97.5% and 95.3% recall and 97.4% and 94.7% $F_1$ score, respectively. Rulesets for extracting TB medications generally achieved above 90% recall and precision metrics.

### Binomial proportions estimated from extracted variables

The estimated proportions and their 95% confidence intervals created from the "Yes/No/Not recorded" and "Positive/Negative" extracted variables are given in Table 5.

After accounting for misclassification errors, the proportion of patients with an active TB diagnosis was 18.2% and the proportion of patients with an LTBI diagnosis was 39.7%. 69.7% of patients had performed a tuberculin sensitivity test and 61.9% of all patients had a positive result for the test. The proportions of patients who were ever started on isoniazid, rifampin or B6 were 45.6%, 17.6% and 30.6% percent, respectively.

### Example application of the cohort–association with LTBI treatment

Table 6 presents the results of the regression analysis between demographic characteristics of patients and receipt of LTBI treatment.

**Table 3. Demographics of the patients included in the SMH-TB database, 2011–2018.**

| Variable | Value | Number of patients who attended at least 1 clinic visit (Total N = 3298) | |
|---|---|---|---|
| | | Count | Percentage |
| Age-group in years (median: 45, IQR: 34–58) | 10–20 | 7 | 0.212 |
| | 20–30 | 422 | 12.8 |
| | 30–40 | 802 | 24.3 |
| | 40–50 | 705 | 21.4 |
| | 50–60 | 575 | 17.4 |
| | 60–70 | 388 | 11.8 |
| | 70–80 | 245 | 7.42 |
| | 80–90 | 126 | 3.82 |
| | 90–100 | 30 | 0.910 |
| | 100–110 | 2 | 0.0606 |
| Sex | Female | 1884 | 57.1 |
| | Male | 1417 | 42.9 |
| | Missing[a] | 1 | 0.0303 |
| Born in Canada | Born in Canada | 247 | 7.48 |
| | Born outside Canada | 2619 | 79.3 |
| | Missing[b] | 436 | 13.2 |
| Underhoused[c] | Yes | 80 | 2.42 |
| | No | 3222 | 97.6 |
| Type of health insurance[d] | Ontario Health Insurance Plan (OHIP) | 2859 | 86.6 |
| | Uninsured Person Program (TB-UP) | 221 | 6.69 |
| | Refugee Health Coverage | 78 | 2.36 |
| | University Health Insurance Plan (UHIP) | 41 | 1.24 |
| | Self-payed | 76 | 2.30 |
| | Other[e] | 27 | 0.819 |

IQR: Interquartile range.

[a]May be due to error in data entry at time of patient registration.

[b]Patient dictations did not mention immigration status or country of birth, or no dictations were found.

[c]Underhoused: includes patients living in homeless shelters, group homes or patients with no fixed address.

[d]For patients with more than one type of insurance, only the insurance type used for the latest encounter is displayed in this table.

[e]Includes any patients with an out-of-province insurance, or not recorded insurance type.

## Discussion

The expansion in deployment of EHR throughout hospital and primary care practices over the past decade in high-income countries has established large longitudinal datasets that can be leveraged for a wide range of research and quality improvement purposes. The evolving use of NLP applied to this retrospective data has produced successes in the clinical domain—for example helping to identify and define clinical syndromes, and predict or estimate disease [31–36]. To similarly facilitate research on TB clinical epidemiology, diagnostics, clinical care and program implementation, quality improvement, and linkage for future therapeutics trials and biomarker studies, we developed a retrospective database of TB clinic patients using structured and unstructured EHR data. The cohort and database are unique in the TB community in the transformation of unstructured data into structured variables using natural language rulesets with excellent performance when validated against manual chart abstraction. The rulesets are open access, and the database is accessible for research and open for collaboration with approval from local research ethics board.

**Table 4. Summary of performance metrics on test set for variables extracted from unstructured dictations. Patients included in test set: N = 200.**

| Variable | Recall | Precision | F$_1$ Score |
|---|---|---|---|
| **Demographics** | | | |
| Country of origin | 0.987 | 0.987 | 0.986 |
| Year of immigration | 0.834 | 0.891 | 0.850 |
| Patient is a healthcare worker | 0.850 | 0.897 | 0.871 |
| **Tuberculosis Diagnosis** | | | |
| Known TB exposure | 0.952 | 0.945 | 0.949 |
| BCG vaccination status | 0.852 | 0.887 | 0.859 |
| TST performed | 0.990 | 0.990 | 0.990 |
| TST induration | 0.954 | 0.960 | 0.957 |
| TST interpretation | 0.978 | 0.981 | 0.980 |
| IGRA performed | 1.00 | 1.00 | 1.00 |
| IGRA interpretation | 1.00 | 1.00 | 1.00 |
| Diagnosis of active TB | 0.975 | 0.973 | 0.974 |
| Diagnosis of LTBI | 0.953 | 0.941 | 0.947 |
| **Tuberculosis Medications** | | | |
| Ever started isoniazid | 0.933 | 0.959 | 0.945 |
| Ever started rifampin | 0.962 | 0.974 | 0.967 |
| Ever started pyrazinamide | 0.996 | 0.994 | 0.995 |
| Ever started ethambutol | 0.985 | 0.984 | 0.984 |
| Ever started vitamin B6 | 0.987 | 0.987 | 0.987 |
| **Medical Conditions and Comorbidities**[**] | | | |
| Autoimmune conditions | 0.862 | 0.767 | 0.807 |
| Diabetes | 0.870 | 0.883 | 0.876 |
| Hematological malignancy | 0.748 | 0.748 | 0.748 |
| Non-hematological malignancy | 0.937 | 0.787 | 0.843 |
| Renal failure | 0.807 | 0.849 | 0.827 |
| HIV status | 0.998 | 0.833 | 0.899 |

TB: Tuberculosis; BCG: Bacillus Calmette–Guérin; TST: Tuberculin sensitivity test; IGRA: Interferon gamma release assay; LTBI: Latent tuberculosis infection; HIV: Human immunodeficiency viruses

[**]Patients that had undergone a transplant and patients diagnosed with silicosis were excluded from this table due to having no positive example in the test set.

The strength of the SMH-TB database comes from the inclusion of granular data, achieved by extracting it from unstructured sources using natural language processing. While the database contains standard structured data accessible in a wide variety of EHRs, a large and unique component of our data comes directly from unstructured dictated clinic notes, which contain a vast number of variables that can be used for a broad range of research topics, such as clinical epidemiology and modeling studies. Our regression analysis, provided as an example of the research applicability of the database, reveals that there is a possible association between housing insecurity and LTBI non-treatment, which is a concerning signal that should be reassessed after the extraction of additional data. This finding points to potential inequities in the delivery of care in the SMH TB clinic that have important public health implications and requires further evaluation from a quality improvement, clinical management, and health policy perspective.

The NLP rulesets allow us to create granular patient-level variables from unstructured data accurately and efficiently, reducing the amount of time spent on manual abstraction to a

**Table 5. Binomial proportion estimate and 95% Confidence Interval (CI) using standard binary regression and MC-SIMEX model for binary variables created from extracted variables.** Total patients with at least 1 dictation: N = 3237.

| Description | Count (N = 3237) | Logistic regression estimate (95% CI) | MC-SIMEX model estimate (95% CI) |
|---|---|---|---|
| **Demographics** | | | |
| Healthcare workers | 438 | 13.5% (12.4, 14.8) | 2.48% (2.02, 3.04) |
| **Tuberculosis Diagnosis** | | | |
| Known TB exposure | 706 | 21.8% (20.4, 23.3) | 16.8% (15.3, 18.3) |
| Received BCG vaccination | 1316 | 40.7% (39.0, 42.4) | 24.8% (23.0, 26.7) |
| Performed a TST | 2279 | 70.4% (68.8, 72.0) | 69.7% (68.1, 71.3) |
| Received a positive TST interpretation | 2031 | 62.7% (61.1, 64.4) | 61.9% (60.2, 63.6) |
| Performed an IGRA | 296 | 9.14% (8.20, 10.2) | 9.14% (8.20, 10.2) |
| Received a positive IGRA interpretation | 301 | 9.30% (8.35, 10.3) | 9.30% (8.35, 10.3) |
| Diagnosed with active TB | 640 | 19.8% (18.4, 21.2) | 18.2% (16.8, 19.7) |
| Diagnosed with LTBI | 1473 | 45.5% (43.8, 47.2) | 39.7% (37.8, 41.6) |
| **Tuberculosis Medications** | | | |
| Ever started on isoniazid | 1314 | 40.6% (38.9, 42.3) | 45.6% (43.6, 47.5) |
| Ever started on rifampin | 548 | 16.9% (15.7, 18.3) | 17.6% (16.3, 19.1) |
| Ever started on pyrazinamide | 349 | 10.8% (9.76, 11.9) | 9.99% (8.96, 11.1) |
| Ever started on ethambutol | 348 | 10.8% (9.73, 11.9) | 9.36% (8.32, 10.5) |
| Ever started on vitamin B6 | 986 | 30.5% (28.9, 32.1) | 30.6% (29.0, 32.2) |
| **Medical Conditions and Comorbidities*** | | | |
| Autoimmune conditions | 167 | 5.16% (4.45, 5.98) | 0.259% (0.175, 0.383) |
| Diabetes | 179 | 5.53% (4.79, 6.37) | 0.358% (0.247, 0.517) |
| Hematological malignancy | 71 | 2.19% (1.74, 2.76) | 0.00625% (0.00320, 0.0122) |
| Non-hematological malignancy | 140 | 4.32% (3.68, 5.08) | 0.860% (0.599, 1.23) |
| Renal failure | 65 | 2.01% (1.58, 2.55) | 0.00895% (0.00450, 0.0180) |
| Diagnosed with HIV | 175 | 5.41% (4.68, 6.24) | 5.43% (4.69, 6.26) |
| No relevant medical conditions/comorbidities** | 2569 | 79.4% (77.9, 80.7) | 89.3% (87.9, 90.6) |

MC-SIMEX: Misclassification Simulation Extraction; CI: Confidence interval; TB: Tuberculosis; BCG: Bacillus Calmette–Guérin; TST: Tuberculin sensitivity test; IGRA: Interferon gamma release assay; LTBI: Latent tuberculosis infection; HIV: Human immunodeficiency viruses

*Patients that had undergone a transplant and patients diagnosed with silicosis were excluded from this table due to having no positive example in the test set.

**Includes any patient with an extracted value of "No/Not recorded/Negative" for all medical conditions/comorbidities listed in the table.

**Table 6. Association between demographic characteristics and receipt of LTBI treatment.** Total patients who were diagnosed with LTBI, N = 1473.

| Variable | Value | Received LTBI treatment (N = 828) | | Did not receive LTBI treatment (N = 645) | | OR (95% CI) |
|---|---|---|---|---|---|---|
| | | N | Proportion (95% CI) | N | Proportion (95% CI) | |
| Age-group | 10–40 | 315 | 38.0% (34.1, 42.2) | 242 | 37.5% (33.1, 42.2) | Reference |
| | 40–70 | 449 | 54.2% (50.1, 58.3) | 337 | 52.2% (47.5, 56.9) | 1.02 (0.82, 1.27) |
| | 70–100 | 64 | 7.73% (5.79, 10.3) | 66 | 10.2% (7.72, 13.4) | 0.75 (0.51, 1.09) |
| Sex* | Male | 320 | 38.6% (34.9, 42.5) | 249 | 38.7% (34.5, 43.0) | Reference |
| | Female | 508 | 61.4% (57.5, 65.1) | 395 | 61.3% (57.0, 65.5) | 1.00 (0.81, 1.24) |
| Underhoused | No | 810 | 97.8% (96.4, 98.7) | 621 | 96.3% (94.2, 97.6) | Reference |
| | Yes | 18 | 2.17% (1.29, 3.63) | 24 | 3.72% (2.38, 5.78) | 0.58 (0.31, 1.07) |

LTBI: Latent tuberculosis infection; CI: Confidence interval; OR: Odds ratio.

*Excluding missing.

minimum. Moreover, the large amount of unstructured raw data is a tremendous resource for evaluating and deploying machine learning and deep learning models capable of automatically extracting meaningful variables from clinical notes, and will be an important aspect in future work for improving the database [37–39]. While here we use simple NLP rule-sets, machine learning models and workflows can be developed to leverage the structured and extracted variables for predictive modeling and early warning systems to delineate, for example, those individuals at risk for adverse drug reactions, or the potential for non-compliance of high risk LTBI patients with treatment completion [40–42]. The breadth of data provided makes this a unique and powerful tool in both clinical and computational research. Indeed, the increasingly widespread availability of high quality EHR in healthcare institutions in general, provides progressively abundant clinical datasets for computational research in machine or deep learning models [18].

The main limitations of the SMH-TB database include issues that arise from missing or incorrect data and the limited availability of data for certain variables leading to non-robust natural language rulesets. Data errors can be due to both human and algorithmic mistakes. Much of the burden of including relevant data in clinical dictations lies with the clinician attending the patient and dictating the note. In the absence of a standardized format, as was the case in the SMH TB clinic, variables may not be dictated in a manner that enables their capture by the NLP tools, or are not dictated at all. This was encountered in the SMH TB clinic notes. Addressing this limitation required over 40 hours of consultation between the chart abstractors and computer analysts to refine the rule sets based on the varied dictation styles and phrasing observed during manual chart abstraction. The creation of a shared set of guidelines and standard formatting for TB clinic dictations, containing all variables relevant to the database going forward, will ensure all data required are captured with the planned future database updates.

When the unstructured data undergoes information extraction, mislabeling of variables can occur due to certain rulesets having subpar performance. This issue is especially apparent for variables with scarce availability of labels. For example, in our validation dataset there were no patients with silicosis. The ruleset for classifying silicosis was adapted from other immunosuppressive conditions and expert knowledge in disease. While it is possible that such rulesets are overly confident in assigning a "No" label to patients even if they present with the condition in question, given the rarity of the event in the patient population it was not possible to provide further cases for perfection of refinement of the NLP ruleset. As such, we have indicated the metrics of our variables so that researchers can understand the limitations of the data with which they are working. The binomial analysis was additionally intended to illustrate basic properties of the database, while addressing some misclassification errors. The 200 charts sampled for ruleset refinement were consecutive patients from a set of clinic visits and may not have been sufficient for less common variables such as comorbidities. That is, further ruleset refinement will be needed with additional charts with purposive sampling of true positives of infrequent variables.

## Conclusion

In summary, here we describe the SMH-TB cohort and database which aim to be a resource for scientists who are conducting research into many facets of TB. The database is unique in that it contains highly granular socio-demographic and clinical patient data derived from structured and unstructured EHR data extracted using NLP rulesets. The validated rulesets are provided open access for use and the data base is intended to be available for collaborative studies.

## Acknowledgments

We thank Dr. Natasha Sabur for supporting arbitration for rulesets; Julie Seemangal (TB Outpatient Clinic Co-Lead) and Grace Bezaliel for supporting verification of algorithms to classify patients seen in the TB clinic.

## Author Contributions

**Conceptualization:** Sharmistha Mishra, Jane Batt.

**Data curation:** David Landsman, Ahmed Abdelbasit, Christine Wang, Victoria Pequegnat, Kamalprit Chokar, Sharmistha Mishra, Jane Batt.

**Formal analysis:** David Landsman.

**Funding acquisition:** Sharmistha Mishra, Jane Batt.

**Investigation:** David Landsman, Ahmed Abdelbasit, Christine Wang, Michael Guerzhoy, Ujash Joshi, Shaun Mathew, Chloe Pou-Prom, David Dai, Joshua Murray, Sharmistha Mishra, Jane Batt.

**Methodology:** David Landsman, Michael Guerzhoy, Ujash Joshi, Shaun Mathew, Chloe Pou-Prom, David Dai, Joshua Murray, Sharmistha Mishra, Jane Batt.

**Project administration:** Michaelia Banning, Muhammad Mamdani, Sharmistha Mishra, Jane Batt.

**Resources:** Victoria Pequegnat, Sharmistha Mishra, Jane Batt.

**Software:** David Landsman, Michael Guerzhoy, Ujash Joshi, Shaun Mathew, Chloe Pou-Prom, David Dai, Joshua Murray.

**Supervision:** Sharmistha Mishra, Jane Batt.

**Validation:** David Landsman, Ahmed Abdelbasit, Christine Wang, Victoria Pequegnat, Sharmistha Mishra, Jane Batt.

**Visualization:** David Landsman, Ahmed Abdelbasit.

**Writing – original draft:** David Landsman, Ahmed Abdelbasit.

**Writing – review & editing:** David Landsman, Ahmed Abdelbasit, Christine Wang, Michael Guerzhoy, Ujash Joshi, Shaun Mathew, Chloe Pou-Prom, David Dai, Victoria Pequegnat, Joshua Murray, Kamalprit Chokar, Michaelia Banning, Muhammad Mamdani, Sharmistha Mishra, Jane Batt.

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
