## [Decision Letter · Decision Letter 0]

27 Oct 2020

PONE-D-20-27001

Cohort profile: St. Michael’s Hospital Tuberculosis Database (SMH-TB), a retrospective cohort of electronic health record data and variables extracted using natural language processing

PLOS ONE

Dear Dr. Batt,

Thank you for submitting your manuscript to PLOS ONE. After careful consideration, we feel that it has merit but does not fully meet PLOS ONE’s publication criteria as it currently stands. Therefore, we invite you to submit a revised version of the manuscript that addresses the points raised during the review process.

We look forward to receiving your revised manuscript.

Kind regards,

Natalia Grabar

Academic Editor

PLOS ONE

Journal Requirements:

Reviewers' comments:

Reviewer's Responses to Questions

**Comments to the Author**

1. Is the manuscript technically sound, and do the data support the conclusions?

Reviewer #1: Yes

Reviewer #2: Partly

2. Has the statistical analysis been performed appropriately and rigorously? 

Reviewer #1: Yes

Reviewer #2: Yes

3. Have the authors made all data underlying the findings in their manuscript fully available?

Reviewer #1: Yes

Reviewer #2: Yes

4. Is the manuscript presented in an intelligible fashion and written in standard English?

Reviewer #1: Yes

Reviewer #2: Yes

5. Review Comments to the Author

Reviewer #1: Overview:

The study details the construction of the SMH-TB database, which the authors claim that could be an important resource for various future research regarding the Tuberculosis condition. Medical knowledge was used to build a set of regular expressions (regex) to extract structured data from clinical narratives, and increase the patients’ variables coverage. The regex ruleset is now available on github, and the SMH-TB could be shared via proper request. The evaluation covered a set of 200 manually annotated patients’ texts, including the calculation of Precision, Recall and F1 metrics.

Main comments:

In general, the article is well written and easy to understand. The study fulfills its objectives and the conclusion is in line with the results achieved. On the other hand, the work does not present any innovative methodology or technique, and has no depth in the use of natural language processing techniques, since it is limited to the use of regular expressions only. The use of some extra pre-processing steps, POS-Tagging algorithms, n-grams, vector representations are examples of techniques that could improve the results, and make the method more generalizable. In addition, there are several systems available to extract clinical concepts from clinical narratives such as cTakes, MetaMap, CLAMP (refer to this study to find more: https://www.sciencedirect.com/science/article/pii/S1532046417301685). The only real novelty is associated with the fact that the authors state that this is the first corpus of its kind available to the scientific community.

More evidence about the quality of the corpus and its usefulness would be important for the study. For example, to test it in some of the tasks associated with tuberculosis research, such as those mentioned in the introduction and discussion (e.g., using the corpus as the gold standard to train a Machine Learning algorithm).

Since the authors used the CHARTextract tool and straightforward techniques, the methodology could be replicated to construct new datasets focusing on diverse diseases and cohorts. Despite requiring, as in this work, the time of annotators with clinical experience for the construction of rules and manual annotation, which sometimes could be the most difficult resource to obtain.

It is not clear for me how the manual labelling occurred. Have you simply marked all the variables (from Table 1) encountered within the text? Have you defined different categories for each entity labeled? Used just exact-match count? A Figure or some examples could easily clarify these questions.

In the “Binomial proportions estimated from extracted variables” section, extra clarifications are needed, like:

a) Which variables were normalized (because some of them are continuous and not categorical)?

b) Using 0/1 binary values could affect, for instance, the use of the corpus for clinical trials screening algorithms? Because a Negative result is very different from Unknown/Not Recorded. Sometimes these values could be the difference between recruiting and not recruiting a patient. So, why not use values that are more granular?

Some text samples could be provided for the reader, so it could be easier to visualize the text extraction challenges, as well as excerpts of the final database.

In conclusion, the article presents the construction of a resource that can be used in future research, however, it presents a shallow methodology of natural language processing for its construction and does not prove its real usefulness and generability through an extrinsic evaluation.

Minor corrections:

mathematical modelling studies >>> mathematical modeling studies

surrounding immigration, housing status, and insurance, and clinical information >>> surrounding immigration, housing status, insurance, and clinical information

co-morbidities >>> comobidities

Consider enhancing Figure 3 quality

Reviewer #2: Dear authors, thank you very much for getting insights into your interesting work, I enjoyed reading it. See below my comments.

TYPE OF SUBMISSION:

Research article.

SUMMARY:

The authors describe the creation of a retrospective clinical data warehouse using structured and unstructured data specifically from tuberculosis outpatients. Rules sets were applied to extract relevant attributes from unstructured narratives and evaluated regarding precision, recall and F-measure. A descriptive analysis was calculated on the final data set.

ESTABLISHMENT OF THE WORK:

The motivation is clear, the paper is well-structured.

The related work is insufficient.

Creating a database using structured and unstructured data from EHRs for specific cohorts has been done. The authors should provide some examples of related work (retrospective cohort building from structured and unstructured data with rules sets).

SUITABILITY OF THE METHODS:

The methods are well described and the rules sets available via GitHub. Precision, recall and F-measure have been applied to the information extraction task, measurements demanded from the clinical NLP community.

APPROPRIATENESS OF THE ANALYSES AND IMPORTANCE OF THE RESULTS:

The analyses are appropriate.

OTHER COMMENTS ABOUT THE PAPER:

Major Revision:

major_001

- Related work

In general such an investigation has not been done the very first time, maybe specifically to Tuberculosis data sets. Please review in detail and improve the related work in this direction.

Minor Revisions:

minor_001

“We developed the first digital retrospective clinical database that combines structured data, unstructured (text) data, and variables derived from transforming unstructured data to structured data using natural language rulesets, among patients assessed in an inner-city outpatient TB clinic at St Michaels Hospital (SMH) of Unity Health Toronto in Toronto, Ontario, Canada.”

the first digital -> a digital

minor_002

Table 3: You can get rid of True Positive*, True Negative* and Accurary. Precision, recall, F-measure is enough. It would be interesting to know how many rules you had to adjust in sum and also per attribute. Have there been some pitfalls when developing them. Shortly discuss limitations of the approach and your experiences. Have there been attributes which were very hard to fetch with a rule set. Have you thought about applying a machine learning based approach for the information extraction task?

minor_003

Are the structured entries further on mapped to a terminology for standardization e.g. SNOMED CT? (Just of interest)

All the best,

Markus

6. PLOS authors have the option to publish the peer review history of their article (what does this mean?). If published, this will include your full peer review and any attached files.

Reviewer #1: No

Reviewer #2: No

---

## [Author Response · Author response to Decision Letter 0]

23 Dec 2020

We thank the editor and reviewers for their careful review of our manuscript and suggestions made for revisions. We have addressed all questions – please find our detailed responses below.

Editor’s Comments

C1. We note that you have indicated that data from this study are available upon request. PLOS only allows data to be available upon request if there are legal or ethical restrictions on sharing data publicly. For information on unacceptable data access restrictions, please see http://journals.plos.org/plosone/s/data-availability#loc-unacceptable-data-access-restrictions.

If there are ethical or legal restrictions on sharing a de-identified data set, please explain them in detail (e.g., data contain potentially identifying or sensitive patient information) and who has imposed them (e.g., an ethics committee). Please also provide contact information for a data access committee, ethics committee, or other institutional body to which data requests may be sent.

R1. There are ethical restrictions on sharing the dataset of unstructured text because unstructured text (dictation notes) cannot be fully de-identified. The hospital institutional ethics board has also placed restrictions on sharing of the de-identified or anonymized dataset of structured variables (and those transformed using the natural language rulesets) without explicit ethics review. The reason is because the structured dataset contains sensitive and comprehensive patient information that when combined, could be identifying. A restriction on open access to, and publication of the dataset has been imposed by the Unity Health Toronto Research Ethics Board, which approved generation of the database. Their letter explaining the restriction, is appended.

However, to facilitate expanded use of (and reproducibility of findings), data requests can be sent to the Data Oversight Committee by contacting the Unity Health Research Ethics Board (researchethics@smh.ca) and the corresponding author. The proposed project for which the data request is being made will be subjected to ethics approval prior to data release. We have provided this information in the Data Availability Statement. 

Reviewers' Comments:

Reviewer #1: 

Main comments:

C1. In general, the article is well written and easy to understand. The study fulfills its objectives and the conclusion is in line with the results achieved. On the other hand, the work does not present any innovative methodology or technique, and has no depth in the use of natural language processing techniques, since it is limited to the use of regular expressions only. The use of some extra pre-processing steps, POS-Tagging algorithms, n-grams, vector representations are examples of techniques that could improve the results, and make the method more generalizable. In addition, there are several systems available to extract clinical concepts from clinical narratives such as cTakes, MetaMap, CLAMP (refer to this study to find more: https://www.sciencedirect.com/science/article/pii/S1532046417301685). The only real novelty is associated with the fact that the authors state that this is the first corpus of its kind available to the scientific community.

R1. We appreciate this suggestion and agree that the unstructured data provide opportunities for novel natural language processing techniques. For this particular project, we used simple rule-sets rather than focus on innovative methodology. We have included in the discussion section that important next steps in database development and improvement include evaluating and deploying more innovative methodologies, including machine learning and deep learning models capable of automatically extracting meaningful variables and concepts from clinical notes (page 19, lines 389-400, tracked manuscript).

C2. More evidence about the quality of the corpus and its usefulness would be important for the study. For example, to test it in some of the tasks associated with tuberculosis research, such as those mentioned in the introduction and discussion (e.g., using the corpus as the gold standard to train a Machine Learning algorithm).

R2. We agree that demonstrating the quality/usefulness of the dataset enhances the manuscript and have completed a logistic regression analysis as a demonstration/example. Specifically, we assessed the association between LTBI treatment vs no treatment and patient housing status, age and sex. (Please see a new Table 6, page 17 and page 19, lines 380-386, tracked manuscript).

C3. Since the authors used the CHARTextract tool and straightforward techniques, the methodology could be replicated to construct new datasets focusing on diverse diseases and cohorts. Despite requiring, as in this work, the time of annotators with clinical experience for the construction of rules and manual annotation, which sometimes could be the most difficult resource to obtain.

R3. We agree that the methodology can be replicated for other datasets in other clinics, but that the resources required for manual annotation are considerable. 

C4. It is not clear for me how the manual labelling occurred. Have you simply marked all the variables (from Table 1) encountered within the text? Have you defined different categories for each entity labeled? Used just exact-match count? A Figure or some examples could easily clarify these questions.

R4. We have generated a new figure (Figure 4, page 9, tracked manuscript) to demonstrate the QuickLabel tool used for the data abstraction and clarified the text (page 9, lines 240 – 258, tracked manuscript) with respect to the manual labelling methodology. The patient’s TST and IGRA status are provided as an example. In brief, manual labelling was completed using the QuickLabel tool which provided both the full text dictation and value labels. The chart abstractors were provided a priori instructions on how to interpret the text to identify the value labels (e.g. Yes, no, indeterminate, not recorded). The abstractor additionally made notes on novel/unanticipated phrasing with which to refine and train the natural language rulesets. Abstractors met with the programmers during iterative refinement of the NLP rulesets on the manually abstracted training database. We show the final dictated text wording and phrasing used for the value labels for the comorbidity variable Diabetes Mellitus, as an example, in a new Table 2 (page 10).

C5. In the “Binomial proportions estimated from extracted variables” section, extra clarifications are needed, like:

a) Which variables were normalized (because some of them are continuous and not categorical)?

b) Using 0/1 binary values could affect, for instance, the use of the corpus for clinical trials screening algorithms? Because a Negative result is very different from Unknown/Not Recorded. Sometimes these values could be the difference between recruiting and not recruiting a patient. So, why not use values that are more granular?

R5. All of the variables used in this section were categorical and thus were not normalized. The categorical variables specified in this section were re-coded to binary (or dichotomous) variables for this analysis only and are still available as categorical variables in the database. The goal of the descriptive analysis in Table 5 (page 15, tracked manuscript), was to illustrate some basic properties of the database, and show how misclassification errors could be incorporated into such an analysis. This is now stated in the text (page 20, lines 422 -423, tracked manuscript).

C6. Some text samples could be provided for the reader, so it could be easier to visualize the text extraction challenges, as well as excerpts of the final database.

R6. An example of the specific text samples encountered in the dictated clinic notes, and used to define the value labels, are shown in the new table 2 (page 10, tracked manuscript). Forty-eight different dictation phrasings were identified to indicate the presence of Diabetes Mellitus. Thirty-eight different phrases identified the absence of Diabetes Mellitus. 

Figure 4 (page 9, tracked manuscript) also addresses this by showing the final variable values for the example text. The corresponding sentence that contains the text is highlighted in yellow. 

C7. In conclusion, the article presents the construction of a resource that can be used in future research, however, it presents a shallow methodology of natural language processing for its construction and does not prove its real usefulness and generability through an extrinsic evaluation.

R7. We trust that the additional data provided in the manuscript, in addition to the regression analysis reinforces the usefulness of this database for future TB research and QI initiatives in TB clinical care.

C8. Minor corrections:

mathematical modelling studies >>> mathematical modeling studies

surrounding immigration, housing status, and insurance, and clinical information >>> surrounding immigration, housing status, insurance, and clinical information

co-morbidities >>> comorbidities

R8. These grammatical and spelling errors are corrected

C9. Consider enhancing Figure 3 quality

R9. This is completed. In addition all figures have now bee run through “PACE” online as per PLoS One requirements to ensure adequate quality of the tiff figures.

Reviewer #2: 

C1 The motivation is clear, the paper is well-structured. The related work is insufficient. Creating a database using structured and unstructured data from EHRs for specific cohorts has been done. The authors should provide some examples of related work (retrospective cohort building from structured and unstructured data with rules sets).

R1. In our introduction and discussion (page 4, lines 121-126 and page 18, lines 362 – 372, tracked manuscript) we now bring forward related studies for discussion where longitudinal clinical databases have been generated from both structured and unstructured data.

C2 .In general such an investigation has not been done the very first time, maybe specifically to Tuberculosis data sets. Please review in detail and improve the related work in this direction.

R2. Thank you for the suggestion, and we agree that our focus was on providing details of the database as developed using Tuberculosis clinical data. We did not pursue innovative or new methodologies, which comprise important next steps. We have revised the discussion to address this point (page 19, lines 389 – 400, tracked manuscript). In the meantime, we demonstrate using an example of a simple regression analysis (new Table 6) – the potential utility of the database for future clinical research.

C3. Minor Revisions:

minor_001. “We developed the first digital retrospective clinical database that combines structured data, unstructured (text) data, and variables derived from transforming unstructured data to structured data using natural language rulesets, among patients assessed in an inner-city outpatient TB clinic at St Michaels Hospital (SMH) of Unity Health Toronto in Toronto, Ontario, Canada.”

the first digital -> a digital

R3. We have made the change on page 5, line 127 of the tracked manuscript.

C4. Table 3: You can get rid of True Positive*, True Negative* and Accuracy. Precision, recall, F-measure is enough. It would be interesting to know how many rules you had to adjust in sum and also per attribute. Have there been some pitfalls when developing them. Shortly discuss limitations of the approach and your experiences. Have there been attributes which were very hard to fetch with a rule set. Have you thought about applying a machine learning based approach for the information extraction task?

R4. Thank you for this suggestion. We agree that these measures may be biased particularly when we have very few positive labels. We have removed the three mentioned columns from the renumbered Table 4 (page 14, tracked manuscript).

In excess of 40 hours was spent in meetings between chart extractors and computer analysts reworking the rulesets based on the observations of varied dictation styles made during the manual chart abstraction. Although we did not log the hours spent on each variable, certainly some variables created more problems than others. For example, essentially no adjustments were required for “date of immigration” or “country of origin”. In contrast, distilling the language used to define the presence or absence of the various co-morbidities, was challenging and required iterative development. The example of the final language defining the value labels for the co-morbidity diabetes mellitus in now shown in Table 2 (page 10, tracked manuscript). We also include a discussion of these problems in the manuscript (page 10, lines 251-258 and page 19, lines 403-412, tracked manuscript).

Important next steps for this work include applying machine learning based approach to information extraction, and we include this in the discussion (page 19, lines 389-392, tracked manuscript). 

C5. Are the structured entries further on mapped to a terminology for standardization e.g. SNOMED CT? (Just of interest)

R5. At the present time we have not mapped the structured data to a standard such as SNOMED CT, but we plan to explore this in the future.

---

## [Decision Letter · Decision Letter 1]

16 Feb 2021

Cohort profile: St. Michael’s Hospital Tuberculosis Database (SMH-TB), a retrospective cohort of electronic health record data and variables extracted using natural language processing

PONE-D-20-27001R1

Dear Dr. Batt,

We’re pleased to inform you that your manuscript has been judged scientifically suitable for publication and will be formally accepted for publication once it meets all outstanding technical requirements.

Kind regards,

Natalia Grabar

Academic Editor

PLOS ONE

Reviewers' comments:

Reviewer's Responses to Questions

**Comments to the Author**

1. If the authors have adequately addressed your comments raised in a previous round of review and you feel that this manuscript is now acceptable for publication, you may indicate that here to bypass the “Comments to the Author” section, enter your conflict of interest statement in the “Confidential to Editor” section, and submit your "Accept" recommendation.

Reviewer #1: All comments have been addressed

Reviewer #2: All comments have been addressed

2. Is the manuscript technically sound, and do the data support the conclusions?

Reviewer #1: Yes

Reviewer #2: Yes

3. Has the statistical analysis been performed appropriately and rigorously? 

Reviewer #1: Yes

Reviewer #2: Yes

4. Have the authors made all data underlying the findings in their manuscript fully available?

Reviewer #1: Yes

Reviewer #2: Yes

5. Is the manuscript presented in an intelligible fashion and written in standard English?

Reviewer #1: Yes

Reviewer #2: Yes

6. Review Comments to the Author

Reviewer #1: All comments and questions raised by me have been addressed in some way in the article.

Although the work continues to rely on straightforward NLP approaches, the authors make this clear in the discussion section.

An additional experiment using a regression algorithm has been added, making the claim that the corpus can be useful for other tasks more robust.

A new figure has been added to make it clear to the reader how they've performed the manual annotation process. In addition, more examples of textual excerpts were presented throughout the article.

Therefore, I consider resolved the issues raised by me.

Reviewer #2: Dear authors, thank you very much for getting insights into your interesting work, I enjoyed again reading it. See below my comments.

OTHER COMMENTS ABOUT THE PAPER:

Thank you for your detailed response to my comments. You clearly considered them in the current version of the paper. Overall in sum with the other reviewer suggestions the quality of the manuscript clearly approved to a level ready for publishing.

7. PLOS authors have the option to publish the peer review history of their article (what does this mean?). If published, this will include your full peer review and any attached files.

Reviewer #1: No

Reviewer #2: No

---

## [Editor Report · Acceptance letter]

22 Feb 2021

PONE-D-20-27001R1 

Cohort profile: St. Michael’s Hospital Tuberculosis Database (SMH-TB), a retrospective cohort of electronic health record data and variables extracted using natural language processing 

Dear Dr. Batt:

I'm pleased to inform you that your manuscript has been deemed suitable for publication in PLOS ONE. Congratulations! Your manuscript is now with our production department. 

Kind regards, 

on behalf of

Dr. Natalia Grabar 

Academic Editor

PLOS ONE